# Daily Behaviors, Worries and Emotions in Children and Adolescents with ADHD and Learning Difficulties during the COVID-19 Pandemic

**DOI:** 10.3390/children8110995

**Published:** 2021-11-02

**Authors:** Terpsichori Korpa, Theodora Pappa, Giorgos Chouliaras, Anastasia Sfinari, Anna Eleftheriades, Matthaios Katsounas, Christina Kanaka-Gantenbein, Panagiota Pervanidou

**Affiliations:** 1Unit of ADHD and Learning Disorders, Department of Child and Adolescent Psychiatry, “P.&A. Kyriakou” Children’s Hospital, 115 27 Athens, Greece; terpsikorpa@gmail.com (T.K.); mkatsounas@gmail.com (M.K.); 2First Department of Pediatrics, School of Medicine, NKUA, “Aghia Sophia” Children’s Hospital, 115 27 Athens, Greece; dorpap@hotmail.com (T.P.); chriskan@med.uoa.gr (C.K.-G.); 3Second Department of Pediatrics, School of Medicine, NKUA, “P.&A. Kyriakou” Children’s Hospital, 115 27 Athens, Greece; georgehouliaras@msn.com; 4Unit of Developmental and Behavioral Pediatrics, First Department of Pediatrics, School of Medicine, NKUA, “Aghia Sophia” Children’s Hospital, 115 27 Athens, Greece; anastasiasfin@gmail.com (A.S.); annielefth-28@hotmail.com (A.E.)

**Keywords:** ADHD, learning difficulties, COVID-19, CRISIS, emotion/worries, daily behavior

## Abstract

The aim of the present study was to investigate the effects of the coronavirus crisis on behavioral and emotional parameters in children and adolescents with ADHD and Learning Difficulties. A total of 101 children, 5–18 years old, were included in the study, 63 (44 boys) of which were diagnosed with ADHD and learning difficulties (ADHD/LD) and 38 were healthy children (19 boys). The CRISIS questionnaire for parents/caregivers was used. The questionnaire was completed during the first national lockdown in Greece and the data referred to two time-points: 3 months before, and the past 2 weeks. A significant deterioration in the “Emotion/Worries (EW)” symptoms was observed during the pandemic in the control group (2.62 ± 0.16 vs. 2.83 ± 0.18, *p* < 0.001). No such differences were noted in the ADHD group: 3.08 ± 0.25 vs. 3.12 ± 0.29, *p* = 0.12. Provision of educational and activities support was related to increased EW before the crisis. In ADHD/LD children, higher parental education and child’s younger age were related to increased symptoms of EW. In the entire group, previous mental health conditions, increasing stress due to restrictions, and increased COVID-related worries were positively associated with the EW symptoms during the crisis. Less affected relations with friends and less reduction in contact with people outside the home were negatively related with EW during the crisis. The study revealed specific parameters that negatively affected the emotional and behavioral variables of children with ADHD and learning difficulties.

## 1. Introduction

The COVID-19 pandemic is a global health emergency that generated a public health crisis. During the COVID-19 pandemic, the Greek government decided on a general lockdown which raised concerns regarding the mental health of children and adolescents since confinement as a preventative measure could itself consist of a psychosocial adversity factor that affects families and their children. The effects of the COVID-19 pandemic have been studied in several populations; however, they have not been thoroughly studied among potentially higher-risk groups, such as children and adolescents with attention deficit hyperactivity disorder (ADHD) and learning difficulties (LDs).

ADHD is one of the most common neurodevelopmental disorders in children [1], and it has a negative effect on their development of learning capacity, interpersonal relationships, self-esteem, and emotions [2]. Children and adolescents with ADHD exhibit inattentiveness and higher rates of disengagement on the assigned task, while the hyperactive and impulsive actions often lead to disruptive behaviors [3]. ADHD and learning disorders have been studied throughout the years depicting the frequency of such comorbidity among the population of ADHD children and adolescents [4]. In a review of 17 studies, a mean comorbidity rate of 45.1% was found in students with ADHD, also diagnosed with LDs in writing, reading or mathematics [5]. Research has shown that this population is at higher risk of developing externalizing and internalizing disorders [6]. ADHD and LD adolescents often face challenging social obstacles, which arise from difficulties experienced during their interaction with peers [7,8]. They experience social isolation and self-esteem impairment, which are sustained through adulthood [9,10]. In addition, the aforementioned difficulties are often experienced and worsened during the interaction with other family members or other daily stressful situations [11,12]. A hostile parental behavior resulting from these could also influence the severity of ADHD symptomatology in children and adolescents [13,14,15].

Governments and policymakers around the world chose school closure and home confinement as two necessary measures to restrict the spread of COVID-19 infection. The true impact of school closure on reducing the risk of infection in children and other age groups is still being debated (parents, grandparents and others) [16]. School serves many important functions; it is not only a place for academic learning and social skills development but also a place where students receive emotional support from teachers and classmates. Hence, it is evident that going to school is necessary for the psychological wellbeing of students [17,18]. Due to COVID-19, however, home schooling was established, which led to the isolation of children. This condition was experienced in an involuntary manner and was associated with feelings of loneliness, affecting negatively the children’s mental health [19,20].

Recent studies have proven that the consequences of the pandemic were severe, noting that children were more likely to express negative emotions, and to develop adjustment difficulties and acute stress disorder. In fact, 30% of children that were isolated during the lockdown demonstrated post-traumatic stress disorder symptoms and therefore attended therapeutic sessions in mental health services [21]. Furthermore, children and teenagers reported irritability, fear of epidemic information, distraction, and bonding problems [21]. Children up to 6 years old exhibited severe attention problems [21], whereas children up to 10 years old demonstrated regressive behavior, such as enuresis, poor vocabulary, irritability, continuous mood changes, stress, and changes in their sleeping patterns [22]. Another study showed that this symptomatology persisted after the end of the lockdown [23]. Teenagers experienced emotional worries, depressive feelings, slow cognitive tempo, oppositional defiant, and inattention [24], as well as a severely reduced life quality [25]. Lastly, the amount of time spent worrying about coronavirus and the perceived negative impact was linked to higher levels of mental health issues in children and adolescents [26,27].

The sudden removal of resources during the first national lockdown and the extended isolation could particularly affect this vulnerable group and place it at a higher risk of behavioral exacerbations [28]. Our hypothesis was that the ADHD symptoms of children would be substantially greater than usual when home schooling was implemented, since the impact of chronic stressors is likely to be significantly greater in adolescents with ADHD. Our objective was to investigate the impact of the coronavirus crisis and the related restrictive measures during the first wave of COVID-19 pandemic in Greece on a clinical sample of children and adolescents with ADHD and LDs.

## 2. Materials and Methods

### 2.1. Procedure

Parents and health professionals referred the participating families to this study through the AD/LD Unit, “P.&A. Kyriakou” Children’s Hospital. Parents and caregivers were informed about this study and consent was signed prior to any further study-related task. All children with ADHD/LDs had been previously diagnosed according to standard procedures, as follows: Participants provided all the necessary demographic information before progressing to the screening procedure. All necessary measurements were conducted by registered clinicians of the hospital and parents or caregivers of the sample.

The screening procedure commenced with the evaluation of the potential learning disabilities in children and adolescents of the sample. The respective instruments were delivered in one evaluation day, on which the sample performed the Software for Screening Learning Skills and Difficulties, the Screening of Learning Difficulties for pupils, and the Reading Test—Alpha Test. Finally, parents and caregivers of the sample received the Teacher Report Form questionnaire, to be completed by a child’s teacher before the next appointment in the clinic, approximately 7–10 days later. The screening for ADHD procedure was conducted on a next appointment in the AD/LD clinic, at which the parents and caregivers of the sample responded to the Children Behavior Checklist and the ADHD—Rating Scale-IV. During the same appointment, the K-SADS-PL was conducted, which needed the responses of both the parents/caregivers and the children/adolescents of the sample.

The aforementioned procedure is conducted with all parents and children visiting the AD/LD unit as a standard initial assessment. The participants of this study were families that had contacted the AD/LD unit during the past 6 months.

At the time of this study’s initiation on June of 2020, a diagnostic meeting was conducted, where the experts reviewed the outcomes of the screening procedure and classified the sample in patients with ADHD and learning difficulties, and Controls. During the review of the sample, any patient with comorbid disorders was excluded from the ADHD/LD group of the study to avoid any potential confounding variables that may interfere with the data. Consequently, the scientific team of the AD/LD unit contacted the families of the sample through digital means to inform them about the study, as well as provide the consent form and the CRISIS questionnaire. The data of the questionnaire were sent and received through digital means, whereby the experts of the AD/LD clinic progressed to the analysis of the parents’ and caregivers’ responses.

### 2.2. Participants and Clinical Measures

A total of 101 children (63 ADHD patients, 38 Controls) were included in the analysis. Age was comparable between the two groups: ADHD patients vs. Controls: mean age: 11.4 ± 3.0 (from 9 to 14 years old) vs. 11.4 ± 4.1 (from 8 to 15 years old), *p*-value = 0.97. Families of children with ADHD/LDs were recruited through invitation in the Unit of ADHD and Learning Disorders in Athens Children’s Hospital Aglaia Kyriakou after completing the initial screening procedure as part of the typical services provided by the hospital. Healthy volunteers were recruited from a convenience community sample of children from the area of Athens. The sampling flowchart is presented in Figure 1. Inclusion criteria for the clinical group were: a. an existing clinical diagnosis of ADHD and LDs given at the Unit of ADHD/SDs and according to the standard procedures; b. age of 8–15 years, both sexes; c. ability to speak and write in Greek. Children with chronic physical illnesses, genetic/chromosomal disorders, intellectual disability and/or psychiatric comorbidity were excluded from the study.

### 2.3. Learning Difficulties Screening Procedure

The assessment tools were based on the screening protocol under the Certification of the Greek Ministry of Education, Research and Religious Affairs.

#### 2.3.1. Software for Screening Learning Skills and Difficulties (LAMDA)

The software for screening learning skills and difficulties (LAMDA) is a written and oral speech screening tool for children and adolescents. This tool assesses learning difficulties in the form of games over the aspects of orthography, morphosyntactic processing, oral speech, understanding, vocabulary, working memory, non-verbal mental abilities and music characteristics understanding [29].

#### 2.3.2. Screening of Learning Difficulties for Pupils (SLD)

The Screening of Learning Difficulties (SLD) is designed for children and adolescents of 8 to 15 years of age and assesses written and oral speech, reasoning and mathematics. The administration of this tool is conducted by the expert covering 6 scales in the field of learning difficulties, which refer to speech and language intake and production, reading writing, reasoning, and math exercises [30].

#### 2.3.3. Reading Test—Alpha Test

The Alpha Test is a diagnostic tool used to assess the reading skills of students through multiple exercises that test their performance during the years of mandatory education attendance. This tool assesses decoding, reading morphology, articulation and apprehension [31].

### 2.4. ADHD Screening Procedure

#### 2.4.1. The Child Behavior Checklist (CBCL)–Teacher Report Form (TRF)

These questionnaires are behavior and emotion assessment tools used for screening of internalizing and externalizing problems of children between 6 to 18 years of age. The CBCL is rated by the parents and the TRF is rated by a teacher and they consist of 113 items each. They were designed to investigate six DSM oriented subscales that screen for attention deficit and hyperactivity, affective, oppositional-defiant, anxiety, conduct, and somatic problems [32].

#### 2.4.2. ADHD Rating Scale IV (ADHD–RS–IV)

This screening instrument is used for ADHD symptomatology measurement and is distributed in a parent-rated and a teacher-rated version. It consists of 18 items that investigate the presence and the degree of inattention, as well as impulsive/hyperactive behaviors based on gender and age [33].

#### 2.4.3. Kiddie–Schedule for Affective Disorders and Schizophrenia–Present and Lifetime Version (K–SADS–PL)

K–SADS–PL is a diagnostic instrument that evaluates 32 present and lifetime psychiatric disorders in children and adolescents. During the administration of this semi-structured clinical interview data are obtained from various informants, such as the child or adolescent, the parents and the clinician, who depict the complete portrait of each symptom [34].

### 2.5. The Coronavirus Health Impact Survey (CRISIS)

The Coronavirus health and Impact Survey (CRISIS) identifies predictors of severe and chronic psychopathology [33]. The main predictors of this tool include impairment and disability directly related to the COVID-19 pandemic. More thoroughly, CRISIS was developed to measure behavioral, mental and physical health aspects of an individual’s life based on the influence of the Coronavirus pandemic on an emotional and behavioral level. This instrument investigates the impact of Coronavirus through 6 domains, which include SARS-CoV-2 exposure/infection in the past 2 weeks, COVID worries in the past 2 weeks, life changes due to the pandemic in the past 2 weeks, mood states, substance use, and daily behaviors.

#### Variables Extrapolated from the CRISIS Questionnaire

Nikolaidis et al. [35] showed sufficient internal consistency and good unidimensionality for items in the domains “emotion/worries three months before the crisis”, “emotion/worries during last two weeks” and “COVID worries”. Therefore, items in each of these three domains were summed and the resulting new variables (“EW-b”, “EW-d” and “COVID worries”) were treated as continuous parameters. EW-b, EW-d and COVID worries did not follow the normal distribution according to Shapiro–Wilk test; therefore, they were log-transformed (logEW-b, logEW-d and logCOVID-worries); normality was not rejected for logEW-d and logCOVID-worries, whereas the transformed values of EW-b was rejected. Nevertheless, log-transformed data showed significantly lower skewness (−0.12 compared to 0.39 for raw data) and given the relatively large sample size (101 individuals), parametric procedures were applied. The analysis by Nikolaidis et al. [33] demonstrated that items in “Life changes due to Coronavirus/COVID-19 crisis in the last two weeks” did not fulfil the required criteria of internal consistency and unidimensionality; therefore were included as separate, independent covariates in the analysis. For “daily behaviors” and “use of digital media”, the participating children were classified for each item as worsened behavior or not according to the following: later to bed on weekdays, later to bed on weekends, more sleeping hours on weekdays, more sleeping hours on weekends, fewer days per week of physical exercise, fewer days per week spent outdoors, more time watching television or digital media, more time spent on social media and more time playing videogames. In our data set, regarding the use of substances/tobacco/alcohol, the answers for all participants before and during quarantine was “not at all” and therefore no analysis was feasible. Likewise, regarding the items which are included in the section “coronavirus/COVID-19 health/exposure status”, all children answered “no” to exposure, suspected case, symptoms, positive family members and COVID-related consequences; therefore, items of that particular domain were not eligible for analysis.

## 3. Results

### 3.1. Statistical Analysis

Continuous parameters are presented as mean ± standard deviation, median (interquartile range). Differences of continuous variables between groups were assessed by Student’s *t*-test for independent data. The comparisons between the pre-crisis and during-crisis items in the domains “daily behaviors” and “use of digital media” were performed using the Wilcoxon signed-rank test. Categorical variables are described as absolute (*n*) and relative (%) frequencies and associations were tested by the Fisher’s exact test. Paternal and maternal educational level were summed to form a new continuous parameter named “parental educational level”.

### 3.2. Outcome and Types of Analyses

The principal outcome in the present analysis was EW-d, which describes the psychological deterioration during the quarantine, taking into account the baseline psychological status (i.e., EW-b). The hypothesis was to model EW-d as an outcome of the following three, distinct components: (a) the previous, “baseline”, psychological state, as described by EW-b; (b) the COVID-specific psychological burden as described by COVID worries; and (c) the quarantine-related stressful restrictions/changes in everyday life, as described by the individual items of the domain “Life changes due to Coronavirus/COVID-19 crisis in the last two weeks”. We aimed to assess the described hypothesis by implementing a structural equation model analysis (SEM). Initially, exploratory stepwise, backwards, linear regression analysis was applied on EW-b to identify pre-COVID socioeconomic, demographic and health-related parameters that might be associated with the baseline emotions and worries. Subsequently, a second stepwise, backwards, linear regression analysis on EW-d was implemented, including EW-b, COVID worries, pre-quarantine socioeconomic, demographic and health-related parameters (that might have an additional independent effect on EW-d, i.e., not only through EW-b as a mediator) and individual items of the domain “Life changes due to Coronavirus/COVID-19 crisis in the last two weeks”. Lastly, in the final, reduced model, we tested the interaction between groups and all other significant parameters, as well as the interaction between changes in quality of family relations and stress due to changes in family relations, which are by definition related. Based on the results of the exploratory analysis, a SEM was formulated. Several different measures of goodness-of-fit of the proposed model are available. The overall fit of the model to the observed data is assessed by Chi-squared test against the saturated (full) model. *p*-values > 0.05 support a good fit, with 1 being the optimum. The root mean square error of approximation (RMSEA) measures the discrepancy function obtained by fitting the model to the sample values. A RMSEA ≤ 0.05 accompanied by an upper bound of the 90% CI below 0.05 indicates a close fit of the model. The probability that the computed RMSEA is not significantly over 0.05 is assessed by the measure *p*-close, which should be >0.05 and as close as possible to 1. Residual based diagnostics are the standardized root mean squared residual (SRMR). Values smaller than 0.08 indicate a good fit (0 being the perfect fit). In addition, the coefficient of determination (CDet), analogous to R^2^ in classic linear regression (that is the proportion of the variance of the outcome variable explained by the model) is reported. The best model was considered the one satisfying all the aforementioned criteria. If more than one model fit the data well, parsimony-based criteria (Akaike Information Criterion (AIC) and Bayesian Information Criterion (BIC)) were used to select the best-fitting SEM.

Secondary outcomes were the worsening of items in the domains “daily behaviors” and “use of digital media”, and were assessed using contingency tables and Fisher’s exact test.

Level of statistical significance was set to 0.05. All analyses were performed on a Stata 11.2 MP platform (StataCorp, TX, USA).

### 3.3. Descriptive Statistics

In relation to gender, fewer males were included in the ADHD group compared to Controls: 30.1% vs. 50.0%, *p*-value = 0.058. Demographic, socioeconomic and health-related data are presented in Table 1.

### 3.4. Exploratory Analyses

Emotions/worries showed significant worsening in the Control group but not in ADHD, due to the increased baseline emotions/worries in the latter (pre- vs. during: Controls: 2.62 ± 0.16, (2.57, 2.67) vs. 2.83 ± 0.18, (2.77, 2.89), *p* < 0.001; ADHD: 3.08 ± 0.25, (3.02, 3,14) vs. 3.12 ± 0.29, (3.05, 3.20), *p* = 0.12).

Multivariate regression on EW-b showed that provision of educational support (positive) and parental educational (inversely) were significantly related to the baseline psychological status. In addition, significant interaction terms, with positive effect on EW-b, were ADHD-mental health status, ADHD-number of persons at home and ADHD-provision of activity support (detailed results not shown, available by the authors). Linear regression showed strong positive associations to EW-b and COVID worries, as far as EW-d is concerned. Additionally, the following individual items were statistically significant: stress due to restrictions, change in the relationships with friends, change in contacts and mental health status according to caregiver. Significant interactions were ADHD-parental educational level and ADHD-child’s age. No collinearity between the items of the domain “Life changes due to Coronavirus/COVID-19 crisis in the last two weeks” was detected, supporting the inclusion as independent, individual parameters (detailed results not shown, available from the authors). 

### 3.5. SEM for EW-d

The SEM with the best fit for EW-d is presented in Figure 2 and model specifics (β-coefficients, 95% confidence intervals and *p*-values) in Table 2. In the final model group exercises, a modifying effect on EW-d, through several significant interactions with other parameters, on EW-b and directly on EW-d. The structure fitted the data relatively well according to metrics shown in Table 3. In summary, the EW-b path had a significant positive association with EW-d, as anticipated. The COVID worries path also showed a positive relation burdening the psychological status of the child. The third path included items related to the “Life changes due to Coronavirus/COVID-19 crisis in the last two weeks” domain. Stressful parameters such as restrictions, changes in relations with people and changes in the quality of relations with friends affect negatively the EW-d. The following parameters had ADHD-specific effects on EW-d through significant interaction terms in the final model: Existence of physical problems, child’s age and parental educational level. It is worth mentioning that the caregiver’s perception of the child’s metal health was an independent predictor in the final model.

### 3.6. Changes in Items in the Domains “Daily Behaviours” and “Use of Digital Media”

All items in “daily habits” and “use of digital media” domains showed significant worsening in both groups (all *p*-values < 0.05), with the exception “of sleeping hours during the weekend” (not statistically significant in the ADHD group) and days with physical exercise (not statistically significant in the Control group). Comparisons of the likelihood of worsening in these items, between ADHD patients and controls, are shown in Table 4 In all items, with the exception of physical exercise where no differences were observed, ADHD patients showed significantly lower probability of worsening compared to Controls.

## 4. Discussion

The current study in patients with ADHD and learning difficulties investigated the perceived impact of the COVID-19 pandemic on their daily behaviors, worries and emotions as well as to compare the possible differences with the impact of the crisis on the control group of health volunteers. This is one of the first studies focusing on children with ADHD during the COVID-19 pandemic. In most cases parents reported that the COVID-19 pandemic contributed to worse functioning of their children regarding physical health, media use, and mental health, however, some areas such as sleep during the weekends and physical exercise were not affected.

### 4.1. Emotional and Psychological Effects

The CRISIS questionnaire investigates a variety of behavioral and emotional parameters [35]. Through the analysis emerged a major factor, namely “Emotions/Worries During the last weeks- EW-d”, which was influenced in many ways due to the COVID-19 pandemic. The EW-d was affected mainly by three paths/parameters. The first parameter that affected the EW-d is the previous psychological state of the children which is described as EW-b. The second one is the COVID worries and the third one is the parameter which consists of the items referred to “Life changes due to the Coronavirus/COVID-19 crisis in the last two weeks” [35].

The previous psychological state (EW-b) seems to be affected by various parameters. There is a negative association between the higher educational level of parents and the EW-b. This means that the psychological burden of children tends to increase when the educational level of parents is lower. Furthermore, there is a positive association between the EW-b and the provision of educational support. Previous research confirms these findings, as parents with better educational backgrounds tend to devote more time to childcare than less educated parents [36]. Factors that influence the daily life of parents with lower educational levels, such as family finances, a stressful lifestyle, and worse understanding of children’s needs contribute to the decline of the mental health of children [37].

An important observation was the existence of factors which affected the EW-b positively such as the provision of activities support in ADHD children. Another factor which influenced the EW-b, however, was the increased duration of time spent with other family members due to the implementation of restrictive measures. Most parents were forced to work from home and other young siblings stopped going to school, which in most cases affected the relationship between family members. Beyond doubt, children with ADHD experience interpersonal difficulties and are often described as children that are noisy, talk more than what is normally expected, interrupt others during conversations, are disobedient, and cannot tolerate their emotions [38,39,40]. This sudden change in intra-familial relationships resulting potentially to more disputes and challenges within the household could have possibly imposed a burden on children with ADHD and could be associated with less tolerable behaviors and negative feelings.

In addition, EW-b was positively associated with the worsening parental perceptions of a child’s mental health. As research suggests, children with ADHD could experience greater difficulties when parents are more power assertive and react to their behavior and symptomatology in a less tolerable manner [41]. It is possible that during the quarantine, parents of children with ADHD had a more assertive and authoritative behavior towards their children, potentially leading to the manifestation of ADHD symptomatology and the expression of negative feelings.

The parameter EW-b, which is influenced by the factors mentioned above, seems to have a positive correlation with the EW-d during the lockdown. These findings suggest that the quarantine negatively affected the mental health of children and adolescents with ADHD and LDs. This is in accordance with similar studies, as children were found to experience increased sadness, loneliness, and less enjoyment in daily life during the pandemic [42]. Another study found increased distress, anxiety and depressive symptomatology that negatively affected the mental health functioning of adolescents with ADHD, exhibiting internalizing and externalizing difficulties [43].

The second parameter which affected the EW-d is the COVID worries and it is positively associated with it. COVID worries, referred to worries about the infection from the virus. Previous research showed that COVID worries could be associated with stress related symptoms during the pandemic [44]. Specifically, increased worries regarding safety and health, as well as uncertainty about the future were also present during the period of quarantine [41,42,45]. Adolescents with ADHD experienced elevated levels of coronavirus related distress, depression and anxiety symptoms [46].

The third parameter, “Life changes due to coronavirus/COVID-19 crisis in the last two weeks”, includes items such as restrictions, changes in relations with people and changes in the quality of relations with friends. The implementation of restrictive measures due to COVID-19 seemed to be perceived as stressful and to add a burden on the psychological status of the children. The restrictions could potentially lead to changes in the relationship between children and their families and friends. As expected from other studies, reduction in contacts with people outside the home was positively related with EW during the crisis [44]. In addition, according to other studies [40], our findings confirmed that the condition of restriction could potentially promote conflict within the family environment and therefore worsen the EW-d symptoms. Another reason for this could be the closing of schools due to the COVID-19 regulations. It is possible that the sudden interruption of friendly relationships with classmates could have caused adaptation problems for the children and adolescents with ADHD. Similar results were reported in a study by Lee et al. [47] on children and adolescents during the lockdown. Our findings therefore suggest that the stress caused by the lockdown due to less social contact and changed routine could potentially be associated with poorer mental health.

We also observed the interaction between EW-d and numerous other parameters. The educational level of parents seemed to have affected unexpectedly the psychological state of children during the lockdown. When the parental educational level was higher, the burden added to EW-d seemed to be greater. This is in contrast with the way that the educational level affected the EW-b. This finding is in accordance with research that investigated the effect of parental educational level on COVID-19-related worrying. It was found that parents who hold a bachelor’s or higher degree experienced more severe feelings of fear and stress regarding the epidemic and its threat on their own or family members’ health than parents with less education [48]. This finding suggests that rational management of parental fear, anxiety and negative emotions could be particularly helpful for children with ADHD, since their exposure to these seems to add a burden on their mental health. Another parameter interacting with EW-d was the age of children with ADHD. The increase in age was negatively associated with the EW-d. Older children demonstrated a milder picture of negative emotions and worries. This is consistent with a study of children and adolescents of typical development, showing that children exhibited a greater decline in their emotional wellbeing than adolescents [49]. This finding suggests that older children could potentially have better coping mechanisms than younger ones, allowing them to adapt better to a new challenging environment, as well as that parents of younger children with ADHD should consider building a stronger support network for their family. The final parameter refers to the existence of physical problems in children with ADHD. Children and adolescents reported more physical health problems during quarantine compared to three months prior. This is in accordance with previous studies that also reported an increase in reports of physical health problems during the quarantine [44].

### 4.2. Behavioural Effects

The restrictive measures affected not only the emotional sector but also the daily behaviors of the children. We examined a variety of areas of children’s behavior, such as total hours spent sleeping, the time of going to bed, time spent exercising or time spent on outdoor activities, and the use of digital media. All aspects of “daily habits” and “use of digital media” domains were significantly influenced in both groups. However, there were few areas where functioning was unchanged, on average. “Sleeping hours during the weekend” did not seem to have been influenced in the ADHD group and “Days with physical exercise” did not seem to have been influenced in the Control group. These finings echo previous studies that reported greater screen time during the quarantine as compared to three months prior [40,44]. Our findings were also in accordance with other studies reporting changes in daily life habits [44]. Despite the assumptions that physical exercise would be reduced, research has proven the opposite. In a general population study regarding the physical activity of adolescents and university students it was found that there was an increase in time spent exercising [50]. Another major concern was the greater involvement with the media, resulting potentially to adverse effects in a child’s emotional and physical health. Our study demonstrated that there was an increase in digital media use time, echoing previous research [44]. We, however, concluded that children and adolescents, who experienced increased levels of loneliness and sadness during the quarantine, could have compensated for this through the use of social media. Gaming could also have offered the means for social interaction among young people, when face-to-face interaction was not an option [51]. During periods of crisis, we recommend that families are asked by health care professionals and pediatricians about potential stressors that may lead to the worsening of children’s mental health and the manifestation of new unhealthy behaviors that act as coping mechanisms, such as the extensive use of technology and social media. Parents could benefit from a discussion with a healthcare professional and should express their worries regarding the behavior of their children. An appropriate adaptation of the parental behavior could benefit children with ADHD and LDs. Our study contributed to the further understanding of the impact of chronic stressors such as the COVID-19 pandemic on children and adolescents with ADHD and LDs and provided more data regarding the specific areas where functioning could be changed.

### 4.3. Strengths and Limitations

This study investigated parental perceptions of COVID-19 pandemic-related impact on children with ADHD and LDs compared to healthy volunteers. Parental reports on emotional, psychological and behavioral functioning of their children showed a decline after the implementation of lockdowns and restrictions. The findings of this paper should be considered during school reopening as the behavioral and mental health effects of the COVID-19 crisis presented social and personal crises, daily activities disruptions, familial difficulties, which add on the challenging educational course of children with ADHD and LDs [52]. The academic performance of these children should be considered as a top priority from all educational staff during school reintegration, however, the emergence of behavioral problems might be a challenging issue which will need an immediate response from the education professionals and clinicians.

Our study has several strengths, as we examined a range of issues possibly affecting children with ADHD and LDs. Our sample was relatively large and comprised of a diverse mix of children and adolescents, however, further research is required in order to track the pandemic-related functioning of children with ADHD and LDs over time and particularly as children transition back to school when the restrictions change. More studies following the cohorts of children longitudinally should be conducted to understand the long-term impacts of the COVID-19 pandemic on the functioning of children with ADHD and LDs as well as on their families. In addition, this study utilized a control group, which offered the ability to filter and investigate the findings through the scope of the general population including children of typical development. However, various limitations should be considered. Most of our data mainly derived from parents and their perceptions regarding the behavior and mental health status of children and adolescents, which could potentially be influenced by external parameters, such the parental educational level. Lastly, the recruitment of our sample was conducted through the services of the AD/LD clinic, which therefore resulted in reaching only those families who sought the respective services. Future research should be conducted on the psychosocial effects of the COVID-19 pandemic to children and adolescents of ADHD/LD. Our data signified that even before the pandemic the child’s mental and physical health status of the ADHD/LD Group was significantly lower than the controls. This finding is confirmed by relevant research that showed the psychosocial difficulties of adolescents with ADHD related with school demands and social problems [53]. Samples of higher educational needs should be further investigated on a psychosocial level in terms of the manner that pandemic related restrictions might have positively affected their mental health, as school related demands decrease.

## Figures and Tables

**Figure 1 children-08-00995-f001:**
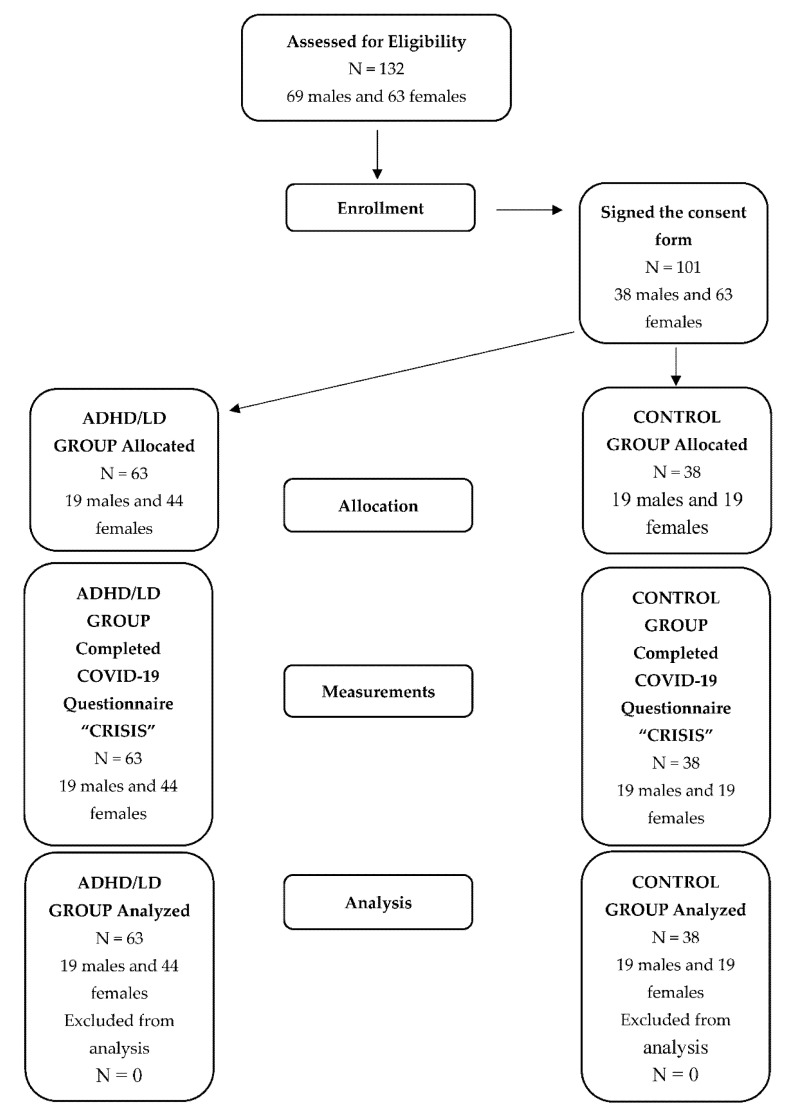
Sampling flowchart.

**Figure 2 children-08-00995-f002:**
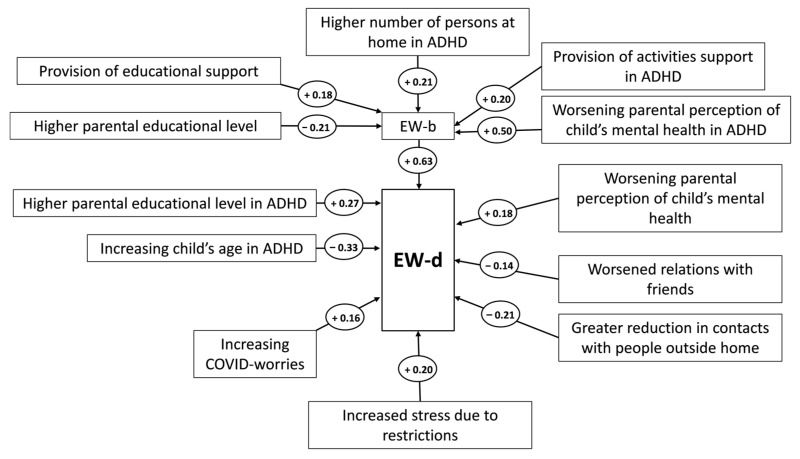
Best-fitting SEM. Effects are represented by connecting arrows with accompanying signs describing the nature of the relation (+ for positive association, − for negative association).

**Table 1 children-08-00995-t001:** Demographic, socioeconomic and health-related data in the study population.

	ADHD	Controls	*p*-Value
Caregiver age, years	44.8 ± 5.4 (40 to 48)	43.4 ± 6.2 (38 to 49)	0.25
Caregiver relation to child			0.10
Mother	46 (73.0)	21 (55.3)	
Father	16 (25.4)	17 (44.7)	
Grandparent	1 (1.6)	0 (0.0)	
Greek origin	62 (98.4)	35 (92.1)	0.15
Living area			0.022
Large city	54 (85.7)	30 (79.0)	
Small city	4 (6.4)	8 (21.0)	
Village	5 (7.9)	0 (0.0)	
Single-parent family	77 (11.1)	6 (15.8)	0.55
Presence of elderly people at home	5 (7.9)	2 (5.3)	0.71
Presence of other children at home	37 (58.7)	27 (71.1)	0.28
Number of other persons at home	2.9 ± 0.7 3 (3, 3)	2.8 ± 0.9, 3 (2, 3)	0.71
Working during quarantine	3 (7.9)	35 (92.1)	<0.001
Working but living in home	29 (46.0)	35 (100.0)	<0.001
Healthcare worker	6 (9.5)	0 (0.0)	0.09
Number of rooms at home	4.4 ± 1.6, 4 (3, 5)	5.4 ± 1.0, 5 (5, 6)	0.001
Insurance	61 (96.8)	38 (100.0)	0.52
Subsidy	14 (22.2)	2 (5.3)	0.026
Child’s physical health status according to caregiver *n* (%)			0.075
Excellent	39 (61.9)	32 (84.2)	
Very good	18 (28.6)	6 (15.8)	
Good	4 (6.3)	0 (0.0)	
Fair	2 (3.2)	0 (0.0)	
Poor	0 (0.0)	0 (0.0)	
Child’s mental health status according to caregiver *n* (%)			<0.001
Excellent	23 (36.5)	34 (89.5)	
Very good	26 (41.3)	4 (10.5)	
Good	12 (19.0)	0 (0.0)	
Fair	2 (3.2)	0 (0.0)	
Poor	0 (0.0)	0 (0.0)	

Continuous variables are presented as mean ± standard deviation, median (interquartile range) and compared by Student’s *t*-test for independent data, whereas categorical parameters are described as absolute (*n*) and relative (%) frequencies and associations were tested by the Fisher’s exact test.

**Table 2 children-08-00995-t002:** SEM * on logEW-d **. Results are presented as β-coefficients, 95% confidence intervals (CI) and *p*-values.

logEW-b ***	β-Coefficient	95% CI	*p*-Value
Interaction ADHD–Parental perception of child’s mental health (worsening)	0.140	0.091, 0.189	<0.001
Increasing parental educational level	−0.030	−0.047, −0.013	<0.001
Interaction ADHD–provision of activities support	0.126	0.032, 0.220	0.009
Educational support, yes vs. no	0.121	0.041, 0.201	0.003
Interaction ADHD-number of persons at home	0.044	0.003, 0.085	0.035
logEW-d			
logEW-b	0.563	0.414, 0.713	<0.001
logCOVID-worries	0.137	0.035, 0.238	0.008
Parental perception of child’s mental health (worsening)	0.065	0.015, 0.115	0.010
Stress due to restrictions	0.059	0.022, 0.095	0.002
Relations with friends (improved)	−0.080	−0.141, −0.019	0.009
Change in contacts (more contacts outside home)	−0.060	−0.093, −0.026	<0.001
Interaction ADHD–reporting of physical health problems	0.102	−0.0004, 0.204	0.051
Interaction ADHD–child’s age	−0.015	−0.025, −0.005	0.003
Interaction ADHD–parental educational level	0.016	0.003, 0.028	0.013

* Structural equation model; ** emotion/worries during COVID crisis; *** emotion/worries before COVID crisis.

**Table 3 children-08-00995-t003:** Measures of goodness-of-fit of the final model presented in Table 2 and Figure 1.

Measure of Goodness-of-Fit	Results
Chi-square test ^a^	13.914 (13), 0.380
RMSEA ^b^	0.027 (<0.001, 0.104), 0.605
SRMR ^c^	0.021
CDet ^d^	0.807

^a^ Compared to the saturated model. Results: value (degrees of freedom), *p*-value; ^b^ Root mean square error of approximation. Results: value (90% CI), pclose; ^c^ Standardized root mean squared residual; ^d^ Coefficient of determination.

**Table 4 children-08-00995-t004:** Differences between ADHD patients and controls, in the likelihood of worsening in items of the domains “daily behaviors” and “use of digital media”.

Item (Probability of Worsening)	ADHD Patients (N = 63)	Controls (N = 38)	*p*-Value
Bedtime weekdays (later)	25 (39.7%)	33 (86.8%)	<0.001
Bedtime weekend (later)	16 (25.4%)	26 (68.4%)	<0.001
Sleeping hours weekdays (more)	17 (27.0%)	21 (55.3%)	0.006
Sleeping hours weekend (more)	5 (7.9%)	12 (31.6%)	0.005
Physical exercise (less)	32 (50.8%)	20 (52.6%)	0.99
Time spent outdoors (less)	24 (38.1%)	32 (84.2%)	<0.001
TV (more)	22 (34.9%)	37 (97.4%)	<0.001
Social media (more)	18 (28.6%)	20 (52.6%)	0.020
Videogames (more)	16 (25.4%)	17 (44.7%)	0.05

Results are presented as absolute numbers and percentages of individuals that showed worsening in each item and compared by Fisher’s exact test.

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
