# Peer review of "Daily Behaviors, Worries and Emotions in Children and Adolescents with ADHD and Learning Difficulties during the COVID-19 Pandemic"

_children, 2021, doi:10.3390/children8110995_

Round 1

Reviewer 1 Report

Improving the development of children and adolescents with ADHD and the on-going management of symptoms for adults with ADHD are important medical and social issues globally.  Therefore, this is an important area for study in the context of the COVID-19 pandemic.   The structure of the text must be revised as the order of the information requires amendment.  The language should be reviewed by a native English speaker who understands this research area or a professional editing service.  There appears to be some misunderstanding over the statistical analysis, so the CIs that include the value zero have been presented as if those results are statistically significant.  It is also unclear how the SEM model was developed and why this was more useful than another type of analysis.  The discussion lacks depth and theory.  It is unclear what this study adds to the scientific knowledge of this subject.  It is also unclear how this might be applied clinically or through education.  This should be stated in a conclusion.  For detailed examples of these issues please see the comments added to the manuscript. 

Author Response

Comment 1:  Improving the development of children and adolescents with ADHD and the ongoing management of symptoms for adults with ADHD are important medical and social issues globally.  Therefore, this is an important area for study in the context of the COVID-19 pandemic.   The structure of the text must be revised as the order of the information requires amendment.  The language should be reviewed by a native English speaker who understands this research area or professional editing service.

"There appears to be some misunderstanding over the statistical analysis, so the CIs that include the value zero have been presented as if those results are statistically significant.    It is also unclear how the SEM model was developed and why this was more useful than another type of analysis".

Responses to Reviewer 1: We thank the Reviewer for his/her insightful comments. The only CI that includes zero is that of “Interaction ADHD-reporting of physical health problems” (-0.0004, 0.204), which is extremely marginal with a p-value of 0.051. The reviewer will agree that statistical significance is an arbitrary cut-off, not an exact science. Therefore, extremely marginal results such as the one discussed should be included in the analysis, since omitting it on the basis of 0.001 would deprive the readers of significant information. Regarding the other CIs, the reviewer has misinterpreted the dash as a separator between the upper and lower limit of the CI. The two limits are separated by commas (,) and the dash denotes negative values.

The structure of the CRISIS questionnaire provides a type of, retrospectively collected, longitudinal data on emotion/worries, on which, information was collected regarding, both, before and during the quarantine. In addition, a series of variables on the effect of different aspects of the restrictions in the psychological status of children, that appeared during the quarantine (and therefore did not exist before) were also recorded (e.g. the effect of restriction on relationships, perception about COVID, etc). It is a logical hypothesis that they all contribute to emotion/worries, the latter being an overall measure of emotion and worries. Given that our aim was to explore which parameters had a significant effect on emotion and worries during the quarantine (considering that, from a clinical point of view, this is a significant concern for pediatricians caring for the general, as well as, special pediatric populations) the described type of data offer the potential to build a hierarchical type of model in time. The SEM analysis offers the tools to decompose a classical regression analysis, revealing mediation effects and complex relations.

Comment 2: 

The discussion lacks depth and theory.  It is unclear what this study adds to the scientific knowledge of this subject.  It is also unclear how this might be applied clinically or through education.  This should be stated in a conclusion.  For detailed examples of these issues please see the comments added to the manuscript. 

Reply: Several additions have been made to the discussion to clarify how the findings may contribute to the existing scientific knowledge and apply to the work of education professionals and clinicians. A conclusion section has been added and all the comments of the reviewer have been attended. 

Reviewer 2 Report

Overall, this is an interesting study and quite a well-written manuscript that has the potential to shed more light on the influence of COVID-19 pandemic on psychological functioning children with ADHD. The aims of the study are original. The group sample size is small. Generally, the introduction and discussion are suitable and wide and new literature was referenced in the appropriate context.

Some points need to be considered:

Introduction:
-    The authors should add an adequate reference to the first paragraph in the "Introduction section".
-    There is a lack of a good theoretical model of clinical psychology about the influence of a specific social situation (e.g., pandemic) on the psychological functioning of children. The authors should describe the adequate theory.

Materials and Methods:
-    The group is small. I have some worries about statistical power and statistical criteria for SEM analysis. Did the authors use the bootstrap procedure? It is not clear.
-    Was sample size calculated (with used e.g., G-Power)? Please explain it.
-    It is not clear. Did the participants have a clinical diagnosis? If not, it is a limitation of the study.
-    There are no inclusion and exclusion criteria for the study.

Results:
-    There is no information on indicators of fit for SEM models. The authors should add this to the Table or Supplementary Materials.

Discussion:
-    There is no "Conclusions section". The authors should add some summarising information to the end "Discussion section".
-    I propose to consider more limitations of the study, e.g., (a) this is no a longitudinal study (the conclusions should be more balanced), (b) this is not an experimental study (rather than a cross-sectional study or correlation study), so it is difficult to interpret results about influence one variable to the second variable.
-    How is the possibility to generalize the results of this study to other populations or populations from another country? Did the authors discuss any cultural factors?
-    In my opinion, the authors should add two important paragraphs: (a) future directions of research and (b) clinical implementations.

Author Response

Overall, this is an interesting study and quite a well-written manuscript that has the potential to shed more light on the influence of COVID-19 pandemic on psychological functioning children with ADHD. The aims of the study are original. The group sample size is small. Generally, the introduction and discussion are suitable and wide and new literature was referenced in the appropriate context.
Some points need to be considered:
Introduction:
-    The authors should add an adequate reference to the first paragraph in the "Introduction section".
-    There is a lack of a good theoretical model of clinical psychology about the influence of a specific social situation (e.g., pandemic) on the psychological functioning of children. The authors should describe the adequate theory.

Response: We thank the Reviewer for his/her insightful comments and suggestions. The Introduction has been extensively revised and the additions suggested by the Reviewer have been made. Please see lines 76 – 107.  Linguistic modifications have been also made to the text.

Materials and Methods:
-    The group is small. I have some worries about statistical power and statistical criteria for SEM analysis. Did the authors use the bootstrap procedure? It is not clear. Was sample size calculated (with used e.g., G-Power)? Please explain it.

Response: The analysis was carried out using Stata. No sample size calculations are available in STATA for SEM models and therefore we cannot provide the actual power for the given sample size. Bootstrap methodology was not used

-    It is not clear. Did the participants have a clinical diagnosis? If not, it is a limitation of the study.
Response: The participants had a clinical diagnosis of ADHD and Learning Difficulties. The clinical sample was derived from the Unit of ADHD and LDs, Department of Child and Adolescent Psychiatry, "P.and A. Kyriacou" Children's Hospital.  Please see lines 125-157.

  •    There are no inclusion and exclusion criteria for the study.
    Response; There are inclusion and exclusion criteria, please see lines 144-147 and 162-174.

  • Results:
    -    There is no information on indicators of fit for SEM models. The authors should add this to the Table or Supplementary Materials.
    Response: We have now added the requested criteria 

  • Discussion:
    -    There is no "Conclusions section". The authors should add some summarising information to the end "Discussion section".
    Response: A conclusion section has been added. Please see lines 656-670.

-    I propose to consider more limitations of the study, e.g., (a) this is no a longitudinal study (the conclusions should be more balanced), (b) this is not an experimental study (rather than a cross-sectional study or correlation study), so it is difficult to interpret results about influence one variable to the second variable.
Response: More limitations have been now added, as suggested by the Reviewer. Please see lines: 630-655.

How is the possibility to generalize the results of this study to other populations or populations from another country? Did the authors discuss any cultural factors?
Response: We discuss findings from other countries, and we do not feel that specific cultural circumstances exist in Greece. Within our sample, there are no subgroups with specific cultural characteristics. 

-    In my opinion, the authors should add two important paragraphs: (a) future directions of research and (b) clinical implementations.
Response: Future research and clinical implementations have been added in the “Strengths and limitations” and in the “Conclusions” sections respectively. Please see lines 628-670.

Round 2

Reviewer 2 Report

The authors have corrected the manuscript after all my suggestions.